# Essential Oils as Post-Harvest Crop Protectants against the Fruit Fly *Drosophila suzukii*: Bioactivity and Organoleptic Profile

**DOI:** 10.3390/insects11080508

**Published:** 2020-08-05

**Authors:** Stefano Bedini, Francesca Cosci, Camilla Tani, Erika Carla Pierattini, Francesca Venturi, Andrea Lucchi, Claudio Ioriatti, Roberta Ascrizzi, Guido Flamini, Giuseppe Ferroni, Isabella Taglieri, Barbara Conti

**Affiliations:** 1Department of Agriculture, Food and Environment, University of Pisa, via Del Borghetto 80, 56124 Pisa, Italy; stefano.bedini@unipi.it (S.B.); francesca.cosci1@virgilio.it (F.C.); camillatani.unipi@gmail.com (C.T.); erika.pierattini@agr.unipi.it (E.C.P.); andrea.lucchi@unipi.it (A.L.); giuseppe.ferroni@unipi.it (G.F.); isabella.taglieri@for.unipi.it (I.T.); 2Technology Transfer Centre—Fondazione Edmund Mach (FEM), Via E. Mach 1, 38010 San Michele all’Adige, TN, Italy; claudio.ioriatti@fmach.it; 3Department of Pharmacy, University of Pisa, Via Bonanno 6, 56126 Pisa, Italy; roberta.ascrizzi@gmail.com (R.A.); guido.flamini@farm.unipi.it (G.F.)

**Keywords:** spotted-wing drosophila, repellence, attractiveness, oviposition deterrence, tea tree, mandarin, organoleptic profile

## Abstract

**Simple Summary:**

The spotted-wing drosophila *Drosophila suzukii* is an invasive small fruit fly that causes extensive damage to many fruit crops. To control this pest, the use of aromatic plants essential oils (EOs) is gaining importance since they are bioactive, biodegradable, and ecologically safe. However, despite of the EOs proved efficacy, they still do not have a widespread application due to their high volatility, composition variability and especially their strong smell. In this study we evaluated not only the EOs bioactivity but also their effects on the organoleptic profile of treated fruits. We tested two EOs extracted from mandarin (*Citrus reticulata*) and tea tree (*Melaleuca alternifolia*) very different for composition and smell. Both the EOs were effective in repelling *D. suzukii*. However, while no negative effects on the organoleptic profiles were detected for the fruits treated with *Citrus reticulata* EO, the fruits treated with *M. alternifolia* EO were defined by the panel of experts as “not suitable for consumption”. Overall, our findings indicate that the use of EOs for the post-harvest protection of small fruits is feasible, provided that the EOs have been selected not only for their bioactivity against the insect pest but also for their affinity with the consumers’ sensorial system.

**Abstract:**

The essential oils extracted from mandarin (*Citrus reticulata* Blanco) fruits, and from tea tree (*Maleleuca alternifolia* (Maiden and Betche) Cheel) leaves have been chemically analyzed and tested for their bioactivity against *D. suzukii*. Besides, to estimate consumers’ acceptability of the essential oil (EO) treatments, we evaluated their impact on the organoleptic characteristics of the EO-treated fruits. The main chemical constituents of the two EOs were 1,8-cineole and 4-terpineol for *M. alternifolia* (22.4% and 17.6% of the total components, respectively), and limonene (83.6% of the total components) for *C. reticulata*. The behavioral tests indicate that the two EOs are able to deter *D. suzukii* oviposition and that *D. suzukii* shows positive chemotaxis to low concentrations of the EOs and negative chemotaxis when the EO concentration increases. While no negative effects on the organoleptic profiles were detected for fruits treated with *C. reticulata* EO, the olfactory profile of fruits treated with *M. alternifolia* EO was so negative that they were defined as “not suitable for consumption” by panellists. Overall, our findings indicate that the use of EOs for the post-harvest protection of small fruits is feasible, provided that the essential oils are selected not only for their bioactivity against the insect pest but also for their affinity with the consumers’ sensorial system.

## 1. Introduction

The spotted-wing drosophila *Drosophila suzukii* (Matsumura) (Diptera: Drosophilidae) is an invasive small fruit fly native to South-East Asia and spread into Western countries, causing extensive damage to many fruit crops. Since its first detection in 2008 in Spain, Italy and California, it has rapidly established in North and South America and in many European countries [1,2,3,4,5]. Its worldwide spread is in part due to the global trade in fresh produce and the cryptic nature of larvae that are hidden inside the fruit, often undetected until after transport [4,6]. The attack of *D. suzukii* is moreover fostered by a number of factors, i.e., widespread cultivation of susceptible crops, distribution of the cultivated land at different altitudes that offers a differentiated and extended fruit ripening period, richness in forests and uncultivated or marginal areas, with the presence of susceptible wild fruits close to the crop plantations [7,8,9].

*D. suzukii* infests a wide range of soft-skinned fruit crops, as well as an ever-growing list of wild fruits [10,11,12,13,14]. The substantial economic losses caused by *D. suzukii* to the fruit industry are primarily due to the oviposition behavior of the species, which chooses ripening fruits as oviposition sites, on the basis of chemical and mechanical cues, unlike other *Drosophila* species, which prefer decaying or rotten fruits [15].

Classic control methods of *D. suzukii* by prophylactic use of synthetic insecticides have been proven to be unsustainable, as the choice of existing products available to growers is limited and also because of pre-harvest interval concerns [4]. Similarly, organic production is seriously threatened because there are few effective organically approved insecticides for *D. suzukii* [16]. For these reasons, integrated pest management (IPM) is considered the best response to control *D. suzukii.* IPM of *D. suzukii* has been carried out in many countries, with chemical and non-chemical strategies [17], i.e., adult monitoring with traps baited with food attractants or synthetic lures [18,19,20], mass trapping techniques [21], push and pull strategy [22], biological, cultural and mechanical control [23,24], and judicious insecticide applications [6,25,26].

Because of the limitations of synthetic pesticides, efforts to control *D. suzukii* are currently focusing on new natural product-based tactics. Different plant products, especially essential oils (EOs), are gaining considerable importance. Despite their shortcomings [27], they are bioactive, biodegradable, and ecologically safe [28,29,30]. EOs are complex mixtures of volatile chemical compounds with multiple modes of action [28]. Several EOs have been tested as fumigant and contact insecticides, oviposition deterrents, attractants and repellents against *D. suzukii* [31,32,33,34,35]. However, despite the large body of evidence for their efficacy, EOs still do not have a widespread application, mainly because of their high volatility, and composition variability but also for their strong smell potentially affecting the organoleptic characteristics of the fruits that may limit their use in food crops.

Mandarin (*Citrus reticulata* Blanco, Rutaceae) and tea tree (*Melaleuca alternifolia* (Maiden and Betche) Cheel, Myrtaceae) EOs had previously been assessed as insect pest control agents, showing significant bioactivity against various insects [36,37,38,39,40]. However, while mandarin EO, extracted from a very common edible fruit, is supposed to have a smell highly compatible with the fruits flavor, the tea tree one, given its herbal smell profile, is supposed to affect negatively the organoleptic properties of EO-treated fruits.

The aims of the present study were to evaluate the bioactivity of the two EOs against *D. suzukii* and to assess their effects on the organoleptic profile of EO-treated fruits, in order to evaluate their suitability as active ingredient in formulations for the fruit post-harvest protection. To do this, we analyzed the chemical composition of the EOs, we assessed their repellence/attractiveness and their effects on the *D. suzukii* oviposition behavior and, finally, we compared their impact on the organoleptic characteristics of the treated fruits.

## 2. Materials and Methods

### 2.1. Mass Rearing of Drosophila suzukii

*D. suzukii* (Matsumura) (Diptera: Drosophilidae) specimens utilized in the experiments (Figure 1A) originated from a colony established in 2010 including around 1000 specimens emerged from infested blueberries and raspberries collected in Valsugana (Trentino, Italy). Wild flies were periodically introduced into the colony in order to minimize genetic drift. *D. suzukii* was reared in Plexiglas^®^ cages (30 × 30 × 30 cm) in laboratory conditions (temperature 22–24 °C, relative humidity 64%, and 14-L: 10-D photoperiod) and was constantly provided with a water wick and artificial diet [41] that served both as a food source and an oviposition medium.

### 2.2. Essential Oils Extraction and Chemical Analysis

*C. reticulata* EO was extracted from pericarps of ripe fruits, while *M. alternifolia* EO was extracted from leaves. Both species were cultivated in the experimental fields of the University of Pisa (Pisa, Italy). The extractions were carried out by hydro-distillation in Clevenger-type apparatus, the extraction time was experimentally determined, and the hydrodistillation was prolonged until no further increase in the EO volume was obtained, for a total of 2 h. The chemical composition of the EOs was assessed by gas chromatography/electron impact mass spectroscopy (GC–EIMS). GC/EIMS analyses were performed with a Varian CP-3800 gas chromatograph equipped with a DB-5 capillary column (30 m × 0.25 mm; coating thickness 0.25 μm) and a Varian Saturn 2000 ion trap mass detector. Analytical conditions were set as in Ascrizzi et al. [42]: injector and transfer line temperatures 220 and 240 °C, respectively; oven temperature programmed from 60 °C to 240 °C, at 3 °C min^−1^; carrier gas helium at 1 mL min^−1^; injection volume 0.2 μL (5% HPLC-grade n-hexane solution); split ratio 1:30. Identification of the constituents was based on comparison of the retention times with those of authentic samples, comparing their Linear Retention Indices (LRIs) with the series of *n*-hydrocarbons, and on computer matching against commercial (NIST 2014 and ADAMS) and a home-made mass spectra library built up from pure substances and components of commercial essential oils of known composition, and MS literature data [43,44].

### 2.3. Behavioral Assays

The repellence/attractiveness of both EOs towards *D. suzukii* adults was evaluated in a two-choice bioassay olfactometer, a Plexiglas unit (15 × 15 × 3 cm) according to Romani et al. [45] with minor changes. In the centre of the unit, a circular chamber (i.e., the specimen release chamber, diameter 4 cm) was connected to two identical chambers by means of two linear paths (length, 2 cm; width, 1 cm), forming a 90° angle. One of the two chambers contained a piece of filter paper (0.8 mm Ø) with 3 µL of *n*-hexane as a control and the other chamber contained the same size filter paper with 3 µL of the 0.25, 0.50, 0.75, 1.00, and 10% of EOs solution in n-hexane, corresponding to 0.60, 1.19, 1.79, 2.39, and 23.87 µL of EO L^−1^ of air of EOs, respectively.

The top of the arena was covered by a removable glass panel. In each trial, at the beginning of the tests, a single individual was gently transferred into the release chamber and carefully released on the floor of the chamber. Every *D. suzukii* adult was observed for 6 min. A choice was judged made if it moved from the central chamber within 20 s after being released and only when it stayed on the chosen source for at least 30 s. For each *D. suzukii*, the dwell time on a given cue was recorded. Individuals that did not make any choice were discarded. With each new specimen, the arena was rotated clockwise 90° to avoid positional effects. Moreover, the relative position of the cues was randomized at each reply. For every bioassay, 30 males and 30 females were tested. After each bioassay, the Plexiglas arena and the glass lid were first washed for about 30 s with hexane, then with warm water at 35–40 °C. The arena was then cleaned in a water bath with mild soap for about 5 min, rinsed with hot water for about 30 s, and finally rinsed with distilled water at room temperature [45].

The oviposition deterrent activity of both EOs was assessed in a two-choice cage, described in Figure 2. Each lateral chamber contained a mock fruit consisting of a 2.5 cm Ø dish of agarized media (10 g) of oviposition substrate composed by 100 mL blueberry juice, 6.5 g fructose, 2.5 g yeast extract, 0.5 g nipagin, 0.5 g benzoic acid and 1.2 g agar. A filter paper disc (Ø 1 cm) was wetted with 100 µL of EOs ethanolic solutions (0.00, 0.01, 0.03, 0.04, 0.06, 0.12, 0.21, 0.50, 1.00, 2.00 and 3.00%, corresponding to 0.00, 0.02, 0.04, 0.06, 0.08, 0.17, 0.3, 0.70, 1.40, 2.80, 4.20 µL L^−1^ air). After ethanol evaporation, the disk was suspended 1 cm above the mock fruit with a paper clip. Five males and five females were released in the central chamber of the cage and the number of eggs laid on the mock fruits in the two lateral chambers was recorded after 24 h (Figure 1B). For each EO, 10 concentrations were tested. Seven replicates of each treatment were performed. A total of 100 µL of ethanol-only treatment was used as a control.

The oviposition stimulation/deterrence effect of the EOs solutions on *D. suzukii* was calculated as follows:OD (%) = (T − C)/(T + C) ∗ 100(1)
where:OD = oviposition deterrence;T = the number of eggs in the treated oviposition substrate;C = the number of eggs in the control oviposition substrate.

Negative values indicate oviposition stimulation; positive values indicate oviposition deterrence.

### 2.4. Organoleptic Characterization of EO-Treated Fruits

Small fruits (strawberries, cherries, and blueberries) originating from organic farming, were purchased from a local large-scale retail market in Pisa (Italy) and treated with *C. reticulata* and *M. alternifolia* EOs as follows: 200 g for each fruit, uniform in size and lacking of imperfections, were washed in running water, air-dried at room temperature, and then dipped (fruits/solution, 1:4 *w/w*) in the EOs water solution (1% EO in a water solution with 1% *v/v* of Tween 80,) for 2 min, by manual stirring [46,47]. Fruits dipped in distilled water only (Control 1) and fruits dipped in 1% Tween 80 water solution (Control 2) were used as controls. All reagents used were of analytical and food grade (Sigma Chemical, Co., St. Louis, MO, USA). After dipping, the fruits were drained on absorbent paper and placed in plastic lidded containers (150 cc, Cuki Cofresco Spa, Turin, Italy) with air as storage atmosphere for 30 min before tasting. Three replicates of each treatment were performed. The organoleptic fruit profiles in terms of smell, taste and touch (rheological features during chewing) of samples as a function of the treatments were evaluated by a panel of 10 trained assessors, 6 females and 4 males, aged between 23 and 60 years (“expert panel”, Department of Agriculture, Food and Environment, University of Pisa). All assessors had previous experience in sensory descriptive analysis, mainly in food and EOs evaluation [48,49]. A specific procedure for the sensory evaluation of EO-treated fruits and the generation of descriptors and their definitions was set up before the tasting sessions. A final set of 19 descriptive parameters, including both quantitative and hedonic attributes, was individuated and evaluated, on a scale of 0–10 (Table 1).

The tasting was carried out in the morning, in a well-ventilated quiet room and in a relaxed atmosphere. To avoid cross contamination, different fruits were assessed in different moments of the same session by the same group of panellists. For each tasting session, each panellist was provided with 3 to 5 fruits depending on their dimensions, without any indication about the dipping treatment. Differently treated samples of each fruit type were assessed separately in the same morning (5 min of wait between two assessing). Furthermore, all treated fruits were removed from the room before a new tasting session started.

The overall organoleptic quality of the EO-treated fruits was expressed as the Overall Hedonic Index (OHI), calculated on the average values attributed during panel tests to each hedonic parameter (fineness of smell, frankness of smell and overall pleasantness—see Table 1), as follows:OHI = Average [Hedonic indexes] × 1.11(2)

The color of the treated fruits was determined using a colorimeter (Eoptis, Mod. CLM-196 Benchtop, Tn., Italy). Fruit color was evaluated by a *CIE L*a*b** color system accepted by the Commission International Eclairage, where *L** is the lightness, *a** and *b** are the red-greenness and blue-yellowness components, respectively. The results were expressed as metric distances among the chromatic coordinates ($\Delta E_{ab}^{*})$ values by the follow equation:(3)$$\Delta E_{ab}^{*}=\sqrt{\Delta L^{*}{}^{2}+\Delta a^{*}{}^{2}+\Delta b^{*}{}^{2}}$$

### 2.5. Statistical Analyses

The proportion of individuals choosing the EO-treated chamber in the two-choice behavioral assays were compared by likelihood chi-square test, with a null hypothesis of a 50:50 chance of insects choosing the control vs. the EO-treated chamber. Oviposition test data were processed by one-way between-groups univariate analysis of covariance (ANCOVA) with the EO as fixed factor. The EO concentration was considered as covariate in the model and its effect was controlled in the analysis. The estimated marginal (EM) means of oviposition deterrence effect are reported. *D. suzukii* behavioral data were processed by SPSS 22.0 software (IBM SPSS Statistics, Armonk, North Castle, NY, USA). Metric distances among the chromatic coordinates data were processed by one-way ANOVA, followed by the Tukey’s b post-hoc test. The reliability of the panel test assessments was analyzed by two-way completely randomized ANOVA with panellists and EO treatments as fixed factors [50,51] with the software Big Sensory Soft 2.0 (ver. 2018, Brescia, Italy), that is specifically developed for sensory analysis.

## 3. Results

### 3.1. EOs Chemical Composition

In *M. alternifolia* EO, 51 chemical constituents were identified, accounting for 96.3% of the entire oil components, while 24 constituents were identified in *C. reticulata* EO, accounting for 99.9% of the whole oil. The most abundant constituent of *M. alternifolia* EO was 1,8-cineole (22.4%), followed by 4-terpineol (17.6%), α-terpineol, δ-cadinene and α-terpinene, whereas limonene (83.6%) was the main chemical compound in *C. reticulata* EO (Table 2). Monoterpene hydrocarbons constituted the most abundant class of compounds in *C. reticulata* EO (90.4%), while oxygenated monoterpenes (45.9%) represented the main chemical class of *M. alternifolia* EO (Table 2).

### 3.2. Behavioral Assays

The tests performed by the two-way olfactometer showed that both EOs exerted a repellent or attractive effect against *D. suzukii* adults, depending on the concentration: at the highest EO tested concentration, a significant repellent effect was observed, while at lower concentrations we observed a significant attractive effect. In general, the repellence increased together with the increment in the EO concentration, and the attractiveness with the decrease in the concentration of the EOs (Figure 3).

The repellence activity was observed at 23.87 mL L^−1^ for *C. reticulata* EO (*χ*^2^ = 6.533, *p* = 0.011), and at 23.87, and 2.39 mL L^−1^ for *M. alternifolia* EO (*χ*^2^ = 19.200, *p* < 0.001; *χ*^2^ = 6.533, *p* = 0.011, respectively) (Figure 3). A significant attractive effect was instead observed at 0.6 and 1.19, mL L^−1^ for both *C. reticulata* (*χ*^2^ = 6.119, *p* = 0.013; *χ*^2^ = 11.267, *p* = 0.001, and *χ*^2^ = 13.067, *p* < 0.001, respectively) and *M. alternifolia* (*χ*^2^ = 17.067, *p* < 0.001; *χ*^2^ = 17.067, *p* < 0.001, and *χ*^2^ = 18.458, *p* < 0.001, respectively) EOs. No significant effect was observed at intermediate concentrations of 2.39 and 1.79 µL L^−1^ (*χ*^2^ = 2.133, *p* = 0.011; *χ*^2^ = 3.379, *p* = 0.069, respectively) for *C. reticulata*, and 1.79 µL L^−1^ (*χ*^2^ = 0.267, *p* = 0.606) for *M. alternifolia* EOs.

In accordance with the results obtained in the two-way olfactometer tests, *C. reticulata* and *M. alternifolia* EOs exhibited different action also on the oviposition activity of *D. suzukii*, depending on the concentration at which they were tested (Figure 4).

Overall, one-way ANCOVA showed no significant difference between the oviposition deterrence activity of *C. reticulata* and *M. alternifolia* EOs after controlling for the effects of the EO dose (F_1, 160_ = 0.553; *p* = 0.458; η_p_^2^ = 0.003) with margin means (estimated at = 0.41 µL L^−1^ air) of 22.17 ± 7.78 and 31.23 ± 9.32 µL L^−1^ air, for *C. reticulata* and *M. alternifolia* EOs, respectively. However, *M. alternifolia* EO showed a stronger correlation between concentration and oviposition deterrence than the *C. reticulata* one (*M. alternifolia* r = 0.790, *p* = 0.006; *C. reticulata* r = 0.622, *p* = 0.055). Moreover, *M. alternifolia* EO showed a complete oviposition deterrence starting from 2.8 µL L^−1^ air; *C. reticulata* EO, instead, was able to exert only 75% of inhibition of the oviposition at the highest dose tested of 4.2 µL L^−1^ air (Figure 4).

### 3.3. Sensorial Analysis of EOs Treated Fruit

With the exception of cherries (Table 3), during the observation period the color of fruits did not change as a consequence of dipping treatment (data not shown). On the other hand, when cherries were dipped in *C. reticulata* or *M. alternifolia* EO solutions, the lightness (*L**) significantly increased in comparison with both control samples, and the *M. alternifolia* EO solution induced a significant color change, also in terms of the increasing of blue-yellow (*b**) components (Table 3), thus indicating a rapid browning of the treated fruits.

Furthermore, when the metric distances among the chromatic coordinates were calculated referring to both controls (Table 4), the cherries dipped in *C. reticulata* showed small color differences, while the color of cherries dipped in *M. alternifolia* EO solution was completely different (6 <$\;\Delta E_{ab}^{*}\;$< 12).

Based on the ANOVA calculated for all quantitative (Table 5) and hedonic (Table 6) parameters evaluated during tasting sessions, differences highlighted for both quantitative and hedonic parameters were significant for most of them, with the formulation of dipping solution recorded as the main effect.

When a new formulation is explored, the level of hedonic quality expressed by the new product is fundamental in determining its consumer acceptability [52]; therefore, despite the panel test being performed by trained judges, some hedonic parameters related to the overall pleasantness and quality of smell were also evaluated (Table 6) to collect some preliminary indications about the organoleptic appeal of the different treatments.

In this context, both controls for each fruit showed the highest values of hedonic parameters in terms of overall pleasantness and frankness, thus indicating that the influence of Tween 80, used for dipping, on the sensorial expression of treated fruits was negligible. On the contrary, the dipping in both the EOs solutions (1%) showed an evident impact on the fruit sensorial expression. The fruits dipped in *M. alternifolia* EO solution were characterized by the worst smell profile, due to the highest number of off-flavors (Figure 5) perceived. Namely, the smell profile developed in the presence of *M. alternifolia* EO was so negative that all the treated fruits were defined as “not suitable for tasting” by panelists (Table 6), determining the lowest values of all the hedonic parameters. On the contrary, the strawberry and blueberry dipped in *C. reticulata* EO maintained an Overall Hedonic Index around the acceptability limit fixed at OHI = 5 (Figure 6).

## 4. Discussion

The use of EOs as insect repellents and oviposition deterrents is of interest, since many of them are allowed as food additives and may be used as a safe mean to protect fruits in both conventional and organic farming. In this work, we observed that both *C. reticulata* and *M. alternifolia* EOs exert a repellent or attractive effect against *D. suzukii* adults, depending on the concentration, and are able to deter oviposition.

In order to get insights about the molecular components of the tested EOs that may act as chemical cues for *D. suzukii* choice, compositions of *C. reticulata* and *M. alternifolia* EOs were characterized by GC–EIMS. In this work, on the basis of the high percentage of limonene (over 80%) detected, we identified the studied *C. reticulata* pericarp as limonene-type according to Fanciullino et al. [53], who reported the existence of two *C. reticulata* peel EO chemotypes: a limonene and a limonene/γ-terpinene one. The limonene one has been reported as the most common *C. reticulata* peel EO chemotype in accessions from the most diverse geographical areas, such as Kenya [54], Egypt [55], Spain [56], as well as in this species native range, identified in China and Japan [57]. Similarly, the existence of different *M. alternifolia* chemotypes is well-established in the literature. Penfold et al. [58] postulated the existence of three *M. alternifolia* chemotypes, distinguished by their 4-terpineol/1,8-cineole relative ratio. Further studies, however, with higher numbers of specimens from different geographical areas, increased the number of possible chemotypes for this species up to five, as the importance of terpinolene as the third key compound in *M. alternifolia* EO was discovered [59]. In 2002, Lee et al. [60] proposed yet another chemotype, counting up to six; finally, Keszei et al. [61] identified a seventh one. These compositional differences have been linked to genetic variations of the terpene synthase, of which four types have been identified so far in *M. alternifolia* [62]. The chemotype variation in *M. alternifolia* natural populations evidences the existence of geographical areas with more “extreme” chemotypes, quantitatively dominated by either 4-terpineol, 1,8-cineole or terpinolene. The *M. alternifolia* EO utilized in the present study exhibited an intermediate chemotype (1,8-cineole/4-terpineol), typical of regions situated at the intersection of the “extreme” chemotype areas [63].

The behavioral and oviposition deterrence tests performed in this work showed that both *C. reticulata* and *M. alternifolia* EOs were able to significantly modify *D. suzukii* behavior. In line with our results, published literature reports *C. reticulata* and *M. alternifolia* EOs repellent activity against *D. suzukii*. In a previous work, Bedini et al. [64] tested the behavior of *D. suzukii* in two-way olfactometer assays, finding that *M. alternifolia* EO showed significant repellence from 1% concentration, while *C. reticulata* EO was repellent only at 10%. *M. alternifolia* EO has been also reported for its repellent properties against the flies *Haemotobia irritans* (Diptera: Muscidae) *Chrysomya megacephala* (Diptera: Calliphoridae) [65]. Similarly, *C. reticulata* EO showed repellent activity against the stored grain insect pests *Sitophilous oryzae* (Coleoptera: Curculionidae), *Tribolium castaneum* (Coleoptera: Tenebrionidae) [66], *Sitophilus zeamais* (Coleoptera: Curculionidae), and *Cryptolestes ferrugineus* (Coleoptera, Cucujidae). On the contrary, an attractive effect of EOs to insects has seldom been reported and, to the best of our knowledge, no reports of the attractiveness to insects of *C. reticulata* and *M. alternifolia* EOs are available.

It is known that olfactory cues play a fundamental role in the choice of oviposition sites for *D. suzukii* [67], and a reduction of *D. suzukii* oviposition rate was detected in field trials using aversive odorants such as octenol [68]. In our experiment, we observed a clear oviposition deterrence on the mock fruits treated by *C. reticulata* and *M. alternifolia* EOs. In agreement with our results, *M. alternifolia* EO (3% concentration) even if belonging to a different chemotype (*M. alternifolia* 4-terpineol chemotype, with a 1,8-cineole relative abundance lower than 2%), completely prevented oviposition of *Lucilia cuprina* (Diptera: Calliphoridae) [36]. Limonene, the main constituent of *C. reticulata* EO, has been found to be a major driver for preference of oviposition in *Drosophila melanogaster* [69]. An effective oviposition deterrence on *D. suzukii* was also found for other EO components. In particular, Renkema et al. [70], in a no-choice assay, observed that thymol, applied on raspberries, reduced female fly landings by 60%, larval infestation by 50% and increased fly mortality compared to controls. On the contrary, Erland et al. [31] observed no significant oviposition deterrent activity for nine EOs from avocado (*Persea americana*), neem (*Azadirachta indica*), kukui nut (*Aleurites moluccana*), macadamia nut (*Macadamia integrifolia*), spike lavender (*Lavandula latifolia*), Grosso lavandin (*Lavandula × intermedia* cv ‘Grosso’), and Provence lavandin (*Lavandula × intermedia* cv ‘Provence’) as well as of the three major monoterpene constituents of lavender EOs (1,8-cineole, 3-carene and linalool), despite their good repellence towards *D. suzukii.*

Regarding the sensorial analysis, *M. alternifolia* EO has a characteristic pungent sensory profile dominated by 1,8-cineole (22.4%) (herbal, medicinal, eucalyptus, mint, camphor) and 4-terpineol (17.6%) (spicy, woody, mint, citrus), while *C. reticulata* EO is predominantly characterized by limonene (83.6%) (fresh, sweet, citrus, orange). Interestingly, both constituents are registered as food additives permitted for direct addition to food for human consumption (US Food and Drug Administration 172.515 Synthetic flavoring substances and adjuvants; 2002/113/EC amending 1999/217/EC as regards the register of flavoring substances used in or on foodstuffs).

Wei and co-workers [46] have reported the efficacy of *M. alternifolia* EO treatment in prolonging the shelf-life of fresh fruits by inhibiting mold growth while preserving their main sensory characteristics. Even if the chemical composition of our *M. alternifolia* EO was different to the one of Wei and co-workers [46], in our experience, the organoleptic profile of fresh fruits was negatively affected by this preserving treatment regardless of the specific fruit considered. As reported before, when *M. alternifolia* EO was used, different off-flavors were detected and, consequently, all fruits were defined as “not suitable for tasting” by panellists.

Furthermore, the *M. alternifolia* EO solution tested in the present work also induced a significant and rapid browning of treated cherries. According to Pathare and co-workers [71], this effect can deeply reduce fruit attractiveness, as the color of a fruit is the first sensory attribute evaluated by consumers since they associate color with flavor, texture and the level of satisfaction.

Overall, although both *C. reticulata* and *M. alternifolia* EOs showed a clear oviposition deterrence activity against *D. suzukii*, the latter significantly altered the sensory quality of the fruit, while *C. reticulata* EO did not. Therefore, according to our results, *C. reticulata* is much more suitable than *M. alternifolia* EO to be used for the formulation of repellents to be applied for the post-harvest protection of small fruits from *D. suzukii* attack.

## 5. Conclusions

The use of botanicals for insect protection is probably as old as crop production itself. However, the combination of the efficacy, speed of action, ease of use, and low cost of synthetic insecticides drove many botanicals to near obscurity in most industrialized countries. However, recently, the many drawbacks of synthetic insecticides led to a resurgence in interest in ‘natural’ means of pest control, and botanicals are again regarded as a new class of ecological products for controlling insect pests. Given their wide spectrum of biological activity against insects, EOs could represent effective, inexpensive, and environmentally friendly alternative pest control agents in conventional and biological agriculture. Furthermore, the progress of biotechnology and nanotechnology might accelerate the emergence of novel and effective EO-based products with long-lasting repellence or attractive effects. However, even if the effectiveness of EOs to control insect pests has been proven, emphasis should be placed on the selection criteria, which should not only be based on their toxicity or repellent activity against the target pest, but also on the organoleptic compatibility with the treated food. Our findings also highlight the fundamental importance of the correct dosage of the EOs in the treatment of fruits. Since the airborne concentration of an EO lessens over time, its effect may switch from repellent to attractive and the EO treatment may increase the oviposition activity of *D. suzukii*. On the other hand, the underestimated attractive activities of EOs deserve further specific studies to evaluate their potential as a lure to be utilized in traps for insect mass trapping.

## Figures and Tables

**Figure 1 insects-11-00508-f001:**
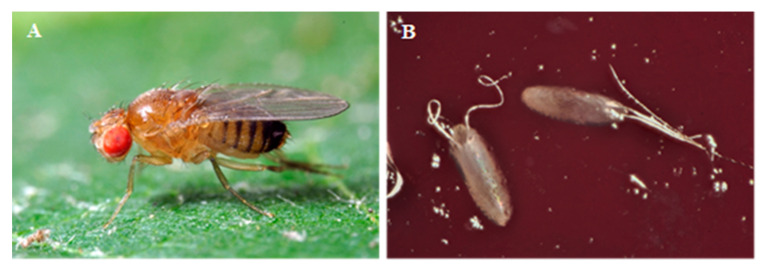
(**A**) Female of *Drosophila suzukii* lateral view. (**B**) *D. suzukii* eggs laid on a mock fruit.

**Figure 2 insects-11-00508-f002:**
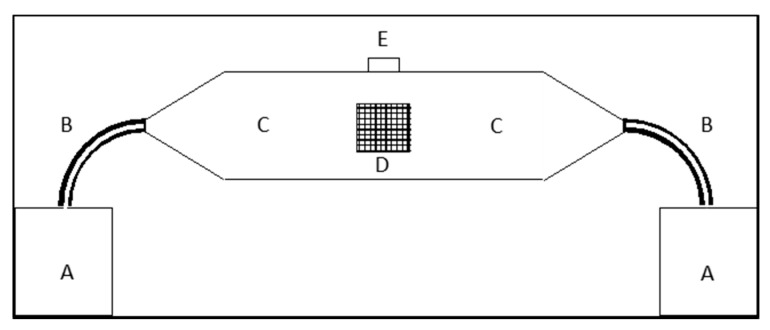
Schematic representation of the two-choice cage used for oviposition deterrence assays. A, 700 mL glass chamber; B, PVC connection tube; C, central chamber; D, net-covered holes for air supply; E, insect entrance (with cap).

**Figure 3 insects-11-00508-f003:**
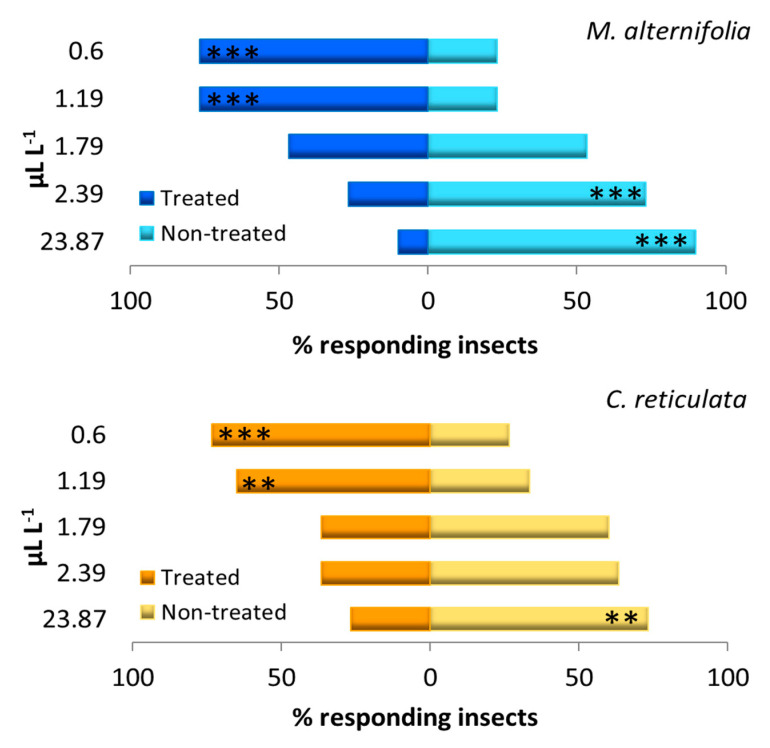
Behavior of adults of *Drosophila suzukii* in the presence of *Melaleuca alternifolia* and *Citrus reticulata* EOs. Histograms represent the percentage of insects that chose the cue or the control chamber. Non-treated, % of insects that chose the control chamber; Treated, % of insects that chose the EO treated chamber. Asterisks indicate significant differences in the number of the choosing insects (*χ*^2^ test; **, *p* < 0.01; ***, *p* < 0.001).

**Figure 4 insects-11-00508-f004:**
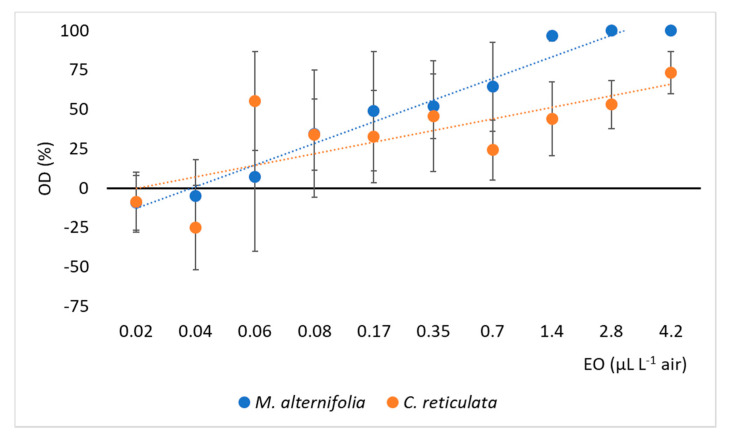
Deterrence of *Melaleuca alternifolia* and *Citrus reticulata* EOs on *Drosophila suzukii* oviposition. OD (%), percentage of oviposition deterrence. Regression equations *M. alternifolia*, y = 13.69x − 26.38; *C. reticulata*, y = 7.35x − 7.69. The bars represent the standard error.

**Figure 5 insects-11-00508-f005:**
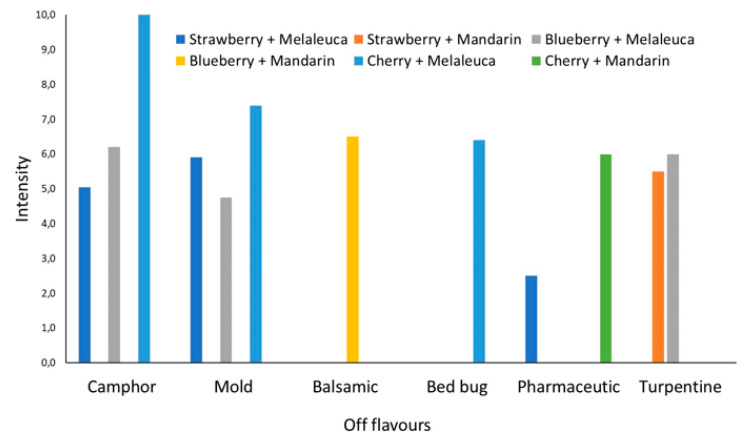
Intensity of off-flavors (camphor, mold, balsamic, bed bug, pharmaceutic, turpentine) attributed by panelists to the essential-oil-dipped fruits (strawberries and blueberries) evaluated on an intensity scale of 0-10. Melaleuca, *Melaleuca alternifolia*; Mandarin, *Citrus reticulata*.

**Figure 6 insects-11-00508-f006:**
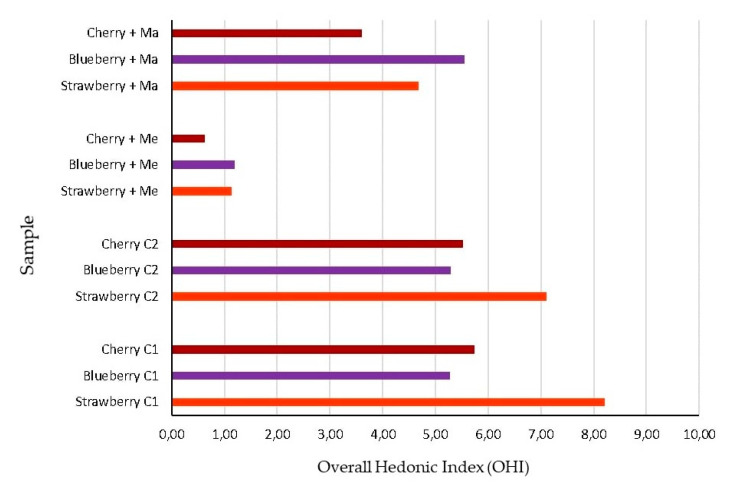
Overall Hedonic Index (OHI) calculated, according to Equation 2, for the essential-oil-dipped fruits (strawberries, blueberries, and cherries) on a scale of 1–10. C1, fruits dipped in distilled water only; C2, fruits dipped in 1% Tween 80 water solution; Me, *Melaleuca alternifolia*; Ma, *Citrus reticulata*.

**Table 1 insects-11-00508-t001:** Descriptive parameters of the organoleptic fruit profiles in terms of smell, taste and touch (rheological features during chewing) ranked by panellists.

Smell	Taste	Touch	Hedonic Parameter
Intensity	Sweet	Hardness	Overall pleasantness
Typical fruit	Acid	Resistance to chewing	Frankness of smell
Floral	Salty	Elasticity	Fineness of smell
Vegetal	Bitter	Springiness	
Spicy		Fibrous behavior	
Citrusy		Granular behavior	

**Table 2 insects-11-00508-t002:** Chemical composition of *Citrus reticulata* and *Melaleuca alternifolia* essential oils (EOs). LRI: linear retention indices on a DB-5 capillary column; -: not detected; tr: traces, <0.1%.

Chemical Compounds	l.r.i.	*Citrus reticulata* EO	*Melaleuca alternifolia* EO
tricyclene	928	0.7	-
α-thujene	931	tr	0.3
α-pinene	941	-	3.2
camphene	954	-	tr
sabinene	976	3.0	-
β-pinene	982	0.1	0.7
myrcene	993	2.4	0.6
octanal	1001	0.8	-
α-phellandrene	1005	-	0.2
α-terpinene	1018	0.1	4.5
*p*-cymene	1027	-	3.1
limonene	1032	83.6	tr
1,8-cineole	1034	-	22.4
*(E)*-β-ocimene	1052	0.3	-
γ-terpinene	1062	0.2	9.0
1-octanol	1071	0.4	-
terpinolene	1089	tr	1.7
linalool	1101	6.0	tr
*cis-p*-menth-2-en-1-ol	1123	tr	0.2
δ-terpineol	1170	-	0.2
4-terpineol	1178	0.6	17.7
*p*-cymen-8-ol	1183	-	tr
α-terpineol	1189	0.5	5.5
*cis*-piperitol	1195	-	tr
decanal	1204	0.5	-
*trans*-piperitol	1207	-	tr
nerol	1230	0.3	-
perilla aldehyde	1273	0.3	-
carvacrol	1298	-	tr
α-cubebene	1350	-	tr
*iso*ledene	1374	-	0.2
α-copaene	1376	tr	0.2
β-patchoulene	1380	-	0.1
α-gurjunene	1410	-	0.9
β-caryophyllene	1420	tr	1.1
β-gurjunene	1432	-	0.2
α-guaiene	1439	-	0.3
aromadendrene	1441	tr	2.9
α-humulene	1456	-	0.3
*allo*aromadendrene	1461	-	1.2
*trans*-cadina-1(6),4-diene	1470	-	0.8
γ-muurolene	1477	-	0.1
β-selinene	1485	-	0.5
δ-selinene	1493	-	0.4
viridiflorene	1494	-	2.0
α-muurolene	1498	-	0.4
δ-cadinene	1524	tr	4.6
cubenene	1534	-	0.5
α-calacorene	1546	-	tr
germacrene B	1556	0.1	-
ledol	1566	-	0.7
spathulenol	1576	-	1.0
globulol	1583	-	3.1
viridiflorol	1590	-	1.1
guaiol	1595	-	1.0
1-*epi*-cubenol	1628	-	1.8
*epi*-α-cadinol	1640	-	0.5
cubenol	1643	-	0.9
selin-11-en-4-α-ol	1653	-	0.2
14-hydroxy-9-*epi-(E)*-caryophyllene	1664	-	0.2
Monoterpene hydrocarbons		90.4	23.2
Oxygenated monoterpenes		7.7	45.9
Sesquiterpene hydrocarbons		0.1	16.8
Oxygenated sesquiterpenes		-	10.4
Non-terpene derivatives		1.7	-
Total identified (%)		99.9	96.3

**Table 3 insects-11-00508-t003:** Chromatic coordinates of the cherries treated with *Citrus reticulata* and *Melaleuca alternifolia* essential oils (EOs).

Sample	*L**	*a**	*b**
Control 1	13.30 a	37.30 a	28.70 a
Control 2	13.50 a	39.00 b	29.07 a
*C. reticulata* EO	17.300 b	39.10 b	30.10 b
*M. alternifolia* EO	22.13 c	37.47 a	35.50 c

Fruits color was evaluated by CIE *L*a*b** color System of *CIE L*a*b*. L**, lightness; *a**, red-greenness; *b**, blue-yellowness. Control 1, fruits dipped in distilled water only; Control 2, fruits dipped in 1% Tween 80 water solution. Different letters indicate significant differences among the treatments according to Tukey’s b post-hoc test (*p* < 0.001).

**Table 4 insects-11-00508-t004:** Color difference among cherries treated with EOs (*Citrus reticulata* and *Melaleuca alternifolia*) and the controls.

	*p*-Value ^1^	Control 1	Control 2	*C. reticulata* EO	*M. alternifolia* EO
Control 1	***	-	1.85 c	4.65 b	11.18 a
Control 2	***	1.85 c	-	4.02 b	10.94 a

Color differences are expressed as metric distances among the chromatic coordinates ($\Delta E_{ab}^{*})$ according to the equation: $\Delta E_{ab}^{*}=\sqrt{\Delta L^{*}{}^{2}+\Delta a^{*}{}^{2}+\Delta b^{*}{}^{2}}$. ^1^ Significance level ***: *p* < 0.001. Control 1, fruits dipped in distilled water only; Control 2, fruits dipped in 1% Tween 80 water solution. For each row, different letters indicate significant differences among the distances according to Tukey’s b post-hoc test.

**Table 5 insects-11-00508-t005:** Quantitative parameters of the sensory analysis ranked by panelists during the tasting of fruits treated with *Citrus reticulata* and *Melaleuca alternifolia* essential oils.

Fruit	Parameter	*p*-Value ^1^	Control 1	Control 2	*M. alternifolia*	*C. reticulata*
Strawberry	Smell attributes					
	Intensity	ns	6.23	6.10	7.45	7.08
	Typical fruit	***	7.92 a	6.03 ab	2.85 b	5.13 ab
	Floral	ns	2.07	1.57	0.33	1.75
	Vegetal	ns	1.00	1.20	0.67	0.83
	Spicy	ns	0.58	0.83	1.42	0.42
	Citrusy	***	0.33 b	0.17 b	0.08 b	5.68 a
	Taste/Touch attributes					
	Sweet	***	3.47 a	4.15 a	n.a.	3.32 a
	Acid	***	6.80 a	5.40 a	n.a.	5.38 a
	Salty	*	1.00 ab	1.92 a	n.a.	1.08 ab
	Bitter	**	1.32 a	1.90 a	n.a.	3.07 a
	Hardness	***	4.00 a	3.37 a	n.a.	2.53 a
	Resistance to chewing	**	3.47 a	3.18 a	n.a.	2.42 ab
	Elasticity	***	1.87 a	1.35 a	n.a.	1.25 a
	Springiness	***	5.33 a	4.83 a	n.a.	4.80 a
	Fibrous behavior	**	2.00 a	1.25 a	n.a.	1.42 a
	Granular behavior	**	2.62 a	1.42 ab	n.a.	2.00 ab
Blueberry	Smell attributes					
	Intensity	*	1.70 b	2.43 b	6.67 a	5.92 a
	Typical fruit	ns	0.33	0.17	0.33	0.17
	Floral	ns	0.00	0.00	0.67	0.67
	Vegetal	ns	0.67	0.35	2.58	1.58
	Spicy	ns	0.00	0.00	1.83	0.50
	Citrusy	***	0.00 b	0.00 b	0.17 b	7.75 a
	Taste/Touch attributes					
	Sweet	***	3.83 a	2.53 a	n.a.	3.92 a
	Acid	***	3.42 ab	5.42 a	n.a.	4.45 a
	Salty	**	0.75 a	1.17 a	n.a.	0.83 a
	Bitter	*	0.72 a	0.55 a	n.a.	2.35 a
	Hardness	***	1.78 a	2.03 a	n.a.	1.70 a
	Resistance to chewing	***	1.37 a	1.75 a	n.a.	1.65 a
	Elasticity	***	2.48 a	2.08 a	n.a.	2.23 a
	Springiness	***	4.78 a	5.62 a	n.a.	5.58 a
	Fibrous behavior	**	0.75 ab	1.17 a	n.a.	0.83 ab
	Granular behavior	ns	0.50	0.67	n.a.	0.57
Cherry	Smell Attributes					
	Intensity	***	1.45 b	1.57 b	7.05 a	5.75 a
	Typical fruit	ns	1.25	1.38	0.00	0.33
	Floral	ns	0.33	0.33	0.00	0.83
	Vegetal	ns	0.75	0.33	1.67	1.00
	Spicy	*	0.00 b	0.00 b	2.17 a	1.33 ab
	Citrusy	***	0.12 b	0.12 b	0.18 b	7.67a
	Taste/Touch attributes					
	Sweet	***	4.67 a	3.92 a	n.a.	3.42 a
	Acid	***	4.37 a	4.87 a	n.a.	6.32 a
	Salty	*	0.83 a	1.08 a	n.a.	0.67 a
	Bitter	***	1.33 b	1.50 b	n.a.	4.90 a
	Hardness	***	3.42 a	3.50 a	n.a.	3.03 a
	Resistance to chewing	***	3.42 a	4.25 a	n.a.	3.02 a
	Elasticity	***	3.15 a	2.92 a	n.a.	3.18 a
	Springiness	***	4.83 a	6.07 a	n.a.	6.27 a
	Fibrous behavior	***	2.50 a	2.75 a	n.a.	2.75 a
	Granular behavior	ns	1.07	1.32	n.a.	1.00

^1^ Significance level. ***: *p* < 0.001 (*F* = 7.10); **: *p* < 0.01 (*F* = 4.43); *: *p* < 0.05 (*F* = 2.87); ns, not significant (*p* > 0.05). ^2^ The ranks of fruits dipped in *M. alternifolia* essential oil solution are not reported as they were defined by the panellists as “not suitable for tasting”. For each row, different letters indicate significant differences among the distances according to Tukey’s b post-hoc test.

**Table 6 insects-11-00508-t006:** Hedonic parameters of the sensory analysis ranked by panelists during the tasting of fruits treated with *Citrus reticulata* and *Melaleuca alternifolia* essential oils.

Fruit	Hedonic Parameters	*p*-Value ^1^	Control 1	Control 2	*M. alternifolia*	*C. reticulata*
Strawberry	Overall pleasantness	***	7.08 a	5.28 a	0.83 c	3.22 b
	Frankness of smell	***	7.97 a	7.33 a	1.12 c	5.17 b
	Fineness of smell	***	7.13 a	6.57 a	1.12 c	4.25 b
Blueberry	Overall pleasantness	***	5.95 a	4.83 a	1.00 b	5.67 a
	Frankness of smell	***	7.58 ab	8.33 a	1.17 c	5.67 b
	Fineness of smell	*	0.72 b	1.13 a	1.05 a	3.67 a
Cherry	Overall pleasantness	***	6.17 a	5.58 a	0.80 c	3.22 b
	Frankness of smell	***	7.67 a	7.58 a	0.62 c	3.57 b
	Fineness of smell	ns	1.65	1.75	0.30	2.97

^1^ Significance level. ***: *p* < 0.001 (*F* = 7.10); *: *p* < 0.05 (*F* = 2.87); ns, not significant (*p* > 0.05). For each row, different letters indicate significant differences among the distances according to Tukey’s b post-hoc test.

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
