# Peer review of "Essential Oils as Post-Harvest Crop Protectants against the Fruit Fly Drosophila suzukii: Bioactivity and Organoleptic Profile"

_insects, 2020, doi:10.3390/insects11080508_

Round 1

Author Response

Reply to Reviewer 1

Many thanks for the time you spent to correct our manuscript. Your corrections and recommendations have been totally performed and have certainly improved the quality of the text.

With regard to Figure 5, the Reviewer is right: there was a mistake in the legend of the figure and we corrected it.

Reviewer 2 Report

Review comments : 867723

Title: Esssential oils as post-harvest crop protectants against the fruit fly Drosophila sujukii

Comment: I suggest changing the title of the article to reflect the scope of the investigation presented in the article. Current title is suggestive that essential oils can be used as crop-protectants against fruit fly, while objectives studied the composition of the essential oil, it’s potential as an repellent and organoleptic characters of fruits treated with essential oil.

Introduction: Authors do not clearly state the objectives of the research. Please state the objectives clearly. More comments on the pdf.

Materials and methods: Please provide details for methods described in lines 128-134. Authors state ovipositional deterrent activity of both EOs and ethanol. Why was ethanol used as only control? Why wasn’t distilled water included as control?
Were both EOs tested at the same time? Or one by one with ethanol as another choice.

Organoleptic characterization:
1. How many panelists were there?

Results:

See comments on pdf.

Author Response

Reply to Reviewer 2

Thank you so much for the time you dedicated to the revision of our manuscript. The changes you suggested have been reported in the text and are listed below.

“I suggest changing the title of this article. The objectives do not address EO as postharvest crop-protectant. A title like this is suggestive of a successful treatment”

R. We changed the title as suggested

Line 18. This sentence is ambiguous. Authors are suggested to clarify…….  

R: We changed the sentence for more clearness as suggested

Line 23. Conclusion sentence is misleading. Authors report that organoleptic……..

R: We changed the sentence in order to better reflect our findings

Introduction: Authors do not clearly state the objectives of the research. Please state the objectives clearly. More comments on the pdf.

R. We changed the “aims” section and stated the objectives of the study more clearly.

Line 45.  Provide reference to support this statement

R. Done

Line 51 e Line 56

R. Done

Line 68. This is a statement not a hypothesis. Consider “EOs with bioactivity against the target pests will have suitable organoleptic characteristics”

R. We changed the “aims” section (see above).

Line 128-134. Please provide details for methods described in lines 128-134. Authors state ovipositional deterrent activity of both EOs and ethanol. Why was ethanol used as only control? Why wasn’t distilled water included as control? 
Were both EOs tested at the same time? Or one by one with ethanol as another choice. 
The experimental procedure needs to be explained in detail. Provide the range of concentration tested

R. We added more detail for the method described (See now lines 250-251). We used ethanol and not distilled water because ethanol was the solvent of the EOs solutions and we wanted to check any effect on the oviposition. The two EOs have been tested one by one in comparison to ethanol as control.

Line 221. X axis in this figure is confusing. Why did the Authors use comma instead decimal point?

R. We changed the commas with points

Line 272. Also report the data for EO treatment.

R. According to the Referee’s suggestion, we have rephrased the sentences (see lines 280-291) and improved the clarity.

Line 281. Define axis in figure 5 and 6. Figures 5 and 6 seem redundant considering M. alternifolia treatment as not suitable for tasting.

R. The axis have been defined. We improved also the clarity in the figure captions. Moreover, we realized that there was a mistake in the legend of the figure 5 and we corrected it.

We preferred to leave both figures 5 and 6 as they provide different information supporting  the results. Figure 5  indeed refers to the off favours attributed by the panelist to the samples after the evaluation of the smell and explains the reason why the fruits dipped in M. alternifolia EO solution were defined by the panellists as “not suitable for tasting”.  Figure 6 shows the Overall Hedonic Index values compared for all the samples, that, although preliminary, gives indications about the organoleptic appeal of the different treatments. As these products are intended for human consumption, it is important  to evaluate the organoleptic compatibility of this kind of treated food together with the effectiveness of the treatment to control insect pest and Figure 6 supports our conclusions.

Line 288. Consider “The use of EOs as insect repellents and oviposition deterrents is of interest as many of them can be allowed as food additives and are safe means to protect fruits in conventional and organic farming”

R. Done

How many panelists were there?

R. The number of panelists was  10, we have added this information in the text

Reviewer 3 Report

MS# Insects-867723

MS title: Essential oils as post-harvest crop protectants against the fruit fly Drosophila suzuki

Brief:

The ms reports on testing of bioactivity of essential oils of mandarin and tea tree against the spotted wing Drosophila (SWD). Besides of providing the chemical characterization of both essential oils and the results against the SWD, the study also reports on the organoleptic profiles of fruits treated with both essential oils. Both oils led to similar protective response towards the pest species, but the tea essential oil compromised the organoleptic attributes of treated fruits preventing its potential utilization.

General comments:

The effort contribution is competent in providing the necessary requirements for such a study exploring SWD response to essential oils. The chemical characterization and the bioassays carried out with the insects were proper and needed. The use of a positive control would have been welcome to anchor the study, but its absence does not impair the appeal as the both EOs tested have already documented biological activity. Furthermore, the addition of the organoleptic study is rather important and frequently neglected, constituting praiseworthy contribution to the study. Thus, my assessment is favorable to the study, and some specific comments/suggestions are detailed below for consideration.

Specific comments:

Title: Clear and to the point; nothing to suggest.

Ln 16: “M. alternifolia” should be in italic;

Ln 47: I would suggest adding a reference here to back up the claim;

Ln 49: The statement truly deserves a reference, as it sounds more like a possibility rather than a reality. Regardless, indicating which is and backing it up is my suggestion;

Ln 56: “limitations” instead of “limits”;

Ln 57: OK, but such products also do have their shortcomings (e.g., Haddi et al. (2020) Pest Manag Sci);

Lns 57-59: Maybe a too broad generalization…?

Lns 59-61: This is also a potential limitation, particularly if the main interacting compounds are not recognized;

Ln 61-62: I truly dislike the term “biocides”, as the last one used in such broad scope are organochlorines, which a way different than modern synthetic and natural compounds under use and scrutiny for use. Please consider replacing the term as it conveys equivocated notion;

Lns 63: “EOs still do not have …”

Lns 104: Identify acronym LRIs at 1st reference in the text;

Lns 151-157: the use of different fruits and double (negative) controls was smart (praiseworthy); a question though – what was the rationale and reason for use of a single concentration of each EOs and I am unsure which concentration was that…? [lns 152-155];

Lns 173: “depending on their dimensions”

Lns 177-183: I have not problem with the use of a general comprehensive parameter or index, such as OHI. However, I believe it to be important to track down and recognize the main contribution parameters for the overall effect observed;

Lns 228: “concentration” rather than “dosage”;

Table 4: These data seem amenable to statistical analyses as different replicates exist and the estimate can likely be made by replicate (and there is independence among them); if so, consider doing it;

Table 5: I would encourage a multivariate analysis with the whole of this quantitative parameters, or better yet the use of a canonical variate analysis (CVA) to secure an overall error level of 5% and subsequently track the main contributors to the overall difference (based on the canonical load of each significant CVA axes);

Fig. 5: unsure of the color code here, as two colors are not represented in the legend (orange and light blue…)

Ln 312: “”extreme” chemotype areas”;

Ln 363: “…, only C. reticulata EO did not significantly…”;

Ln 375: “correct concentration” rather than “correct dosage”;

Author Response

Reply to reviewer 3

Specific comments:

Ln 16: “M. alternifolia” should be in italic;

Done

Ln 47: I would suggest adding a reference here to back up the claim;

Done

Ln 49: The statement truly deserves a reference, as it sounds more like a possibility rather than a reality. Regardless, indicating which is and backing it up is my suggestion;

Done

Ln 56: “limitations” instead of “limits”;

Done

Ln 57: OK, but such products also do have their shortcomings (e.g., Haddi et al. (2020) Pest Manag Sci);

Done. We cited Haddi et al., 2020.

Lns 57-59: Maybe a too broad generalization…?

We delete “particularly for the prevention of post-harvest food loose” and we add “these are bioactive”.

Lns 59-61: This is also a potential limitation, particularly if the main interacting compounds are not recognized;

We delete the sentence “this characteristic enhances their activity due to the synergistic action among constituents”.

Ln 61-62: I truly dislike the term “biocides”, as the last one used in such broad scope are organochlorines, which a way different than modern synthetic and natural compounds under use and scrutiny for use. Please consider replacing the term as it conveys equivocated notion;

We replace the term “biocides” with “insecticides”.

Lns 63: “EOs still do not have …”

Done

Lns 104: Identify acronym LRIs at 1st reference in the text;

Done

Lns 151-157: the use of different fruits and double (negative) controls was smart (praiseworthy); a question though – what was the rationale and reason for use of a single concentration of each EOs and I am unsure which concentration was that…? [lns 152-155];

The EOs concentration used is 1% of EOs in a 1% Tween 80 water solution. We improved the clarity in the text (see line now 194). This is the minimum concentration at which the essential oils showed a statistically significant repellence in olphactometer to the adults of D. suzukii and a percentage of oviposition deterrence around or up 75%.

Lns 173: “depending on their dimensions”

Done

Lns 177-183: I have not problem with the use of a general comprehensive parameter or index, such as OHI. However, I believe it to be important to track down and recognize the main contribution parameters for the overall effect observed;

In Table 1 we reported the hedonic parameters that were evaluated during tasting and that were utilized for the calculation of the Overall hedonic index. The values attributed by panellists to these hedonic indices and utilized for the calculation of the Overall hedonic index were reported in table 6. According to reviewer suggestion, the sentence was rephrased to improve the clarity.

Lns 228: “concentration” rather than “dosage”;

Done

Table 4: These data seem amenable to statistical analyses as different replicates exist and the estimate can likely be made by replicate (and there is independence among them); if so, consider doing it;

According to the Referee’s suggestion, we have changed Table 4 adding the statistical analysis and have modified the text improving the clarity. In this regard, a sentence in Material and methods section was also added.

Table 5: I would encourage a multivariate analysis with the whole of this quantitative parameters, or better yet the use of a canonical variate analysis (CVA) to secure an overall error level of 5% and subsequently track the main contributors to the overall difference (based on the canonical load of each significant CVA axes);

We thank the Referee for the suggestion. We decided however to perform a Two-way ANOVA with panellists and samples as main effect as it is generally accepted as method for the statistical analysis of the results related to sensory characterization by trained panel. Furthermore, our analysis was performed by utilizing a software specifically developed for this purpose, the Big Sensory Soft 2.0 (ver. 2018). We added some references in the Material and Methods section in this regard in order to improve the text.

Fig. 5: unsure of the color code here, as two colors are not represented in the legend (orange and light blue…)

Done. There was a mistake in the legend of the figure 5 and we corrected it. We apologize for that.

Ln 312: “”extreme” chemotype areas”;

Done.

Ln 363: “…, only C. reticulata EO did not significantly…”;

Done, we rephrased the paragraph.

Ln 375: “correct concentration” rather than “correct dosage”;

Done.